# Inspiratory Muscle Training and Its Impact on Weaning Success in Mechanically Ventilated ICU Patients: A Systematic Review

**DOI:** 10.3390/jfmk10020111

**Published:** 2025-03-28

**Authors:** José Luís Alonso-Pérez, Víctor Riquelme-Aguado, Daniel Rodríguez-Prieto, Alejandro López-Mejías, Carlos Romero-Morales, Giacomo Rossettini, Jorge Hugo Villafañe

**Affiliations:** 1Department of Physiotherapy, Faculty of Medicine, Health and Sports, Universidad Europea de Madrid, 28670 Villaviciosa de Odón, Spain; joseluis.alonso@universidadeuropea.es (J.L.A.-P.); danirguezp47@gmail.com (D.R.-P.); alelopezfisio@gmail.com (A.L.-M.); carlos.romero@universidadeuropea.es (C.R.-M.); mail@villafane.it (J.H.V.); 2Department of Basic Health Sciences, Rey Juan Carlos University, 28933 Madrid, Spain

**Keywords:** inspiratory muscle training, mechanical ventilation, weaning success, maximum inspiratory pressure, rapid shallow breathing index, critical care

## Abstract

Background/Objectives: A major importance is now accorded to respiratory muscle weakness resulting from exposure to invasive mechanical ventilation (IMV) in intensive care unit patients. Some authors suggested that Inspiratory Muscle Training (IMT) could increase the chances of weaning off IMV. This systematic review examined the efficacy of IMT on weaning success in mechanically ventilated patients. Methods: A literature search was conducted on PubMed, Cochrane, and PEDro until June 2023. Weaning success, maximum inspiratory pressure (MIP), and Rapid Shallow Breathing Index (RSBI) were the outcome measures included. Results: Seven randomized controlled trials, including 517 participants under IMV for at least 48 h, were included in the review. From a qualitative point of view, a significant increase in MIP and a significant decrease in RSBI were found in the intervention group during the analysis. However, weaning success was the same between the intervention and control groups. No significant association was found between weaning success and the increase of MIP or the decrease of RSBI. Furthermore, it could not be demonstrated that a positive change in MIP or RSBI would increase the weaning success rates. Conclusions: From a qualitative point of view, IMT is effective in increasing MIP and decreasing RSBI. However, IMT has no significant impact on weaning success. Further research is recommended to analyze the effect of IMT on weaning success.

## 1. Introduction

Intensive Care Unit-Acquired Weakness (ICU-AW) is a common and debilitating complication among critically ill patients, characterized by generalized and symmetric muscle dysfunction. Its prevalence is significant, affecting up to 80% of patients in intensive care units (ICUs) and manifesting as critical illness polyneuropathy, myopathy, and pronounced muscle atrophy. These complications contribute to short- and long-term sequelae, negatively impacting patients’ physical health, mental well-being, and overall quality of life [1]. Early and targeted rehabilitation programs are essential to mitigate the progression of ICU-AW. Interventions such as passive mobilization, active physical therapy, and respiratory muscle training have demonstrated efficacy in reducing muscle wasting and improving functional recovery [2,3].

Respiratory muscle weakness is a particularly prevalent issue among ICU patients, reported to be twice as common as limb muscle weakness. This condition is associated with higher extubation failure rates, prolonged dependence on mechanical ventilation, and poor clinical outcomes, including increased hospital and one-year mortality rates [4]. Prolonged periods of invasive mechanical ventilation (IMV) contribute significantly to this weakness, highlighting the importance of efficient and timely weaning processes to restore spontaneous breathing.

Weaning from IMV involves strategies such as Spontaneous Breathing Trials (SBTs) and gradual reductions in ventilator support. SBTs can be performed using techniques such as T-piece trials, Continuous Positive Airway Pressure (CPAP), Pressure Support Ventilation (PSV), or automatic tube compensation [4,5]. These trials typically last between 30 and 120 min, with readiness testing ensuring appropriate timing to minimize the risks of premature weaning or unnecessary ventilation [6]. Parameters such as respiratory rate (RR), tidal volume (VT), the Rapid Shallow Breathing Index (RSBI), and maximum inspiratory pressure (MIP) are crucial for assessing weaning readiness. Optimal thresholds include an RR ≤ 35 breaths/min, VT > 5 mL/kg, RSBI < 105, and MIP values between −20 and −25 cm H_2_O [7,8].

However, weaning failure—defined as failing an SBT or requiring reintubation within 48 h post-extubation—remains a significant clinical challenge. Persistent respiratory muscle weakness and comorbid conditions are frequently implicated, necessitating targeted interventions to improve outcomes [9].

Inspiratory Muscle Training (IMT) has been identified as a potential strategy to enhance weaning success, particularly in patients who encounter difficulties during extubation. IMT focuses on anaerobic training to strengthen the diaphragm and accessory inspiratory muscles. This training employs resistive or threshold loading techniques, with each method offering distinct advantages. Resistive loading allows for variable training intensity based on patient effort, while threshold loading requires the generation of a preset pressure to enable airflow, ensuring consistent and effective muscle engagement [10,11]. IMT has been shown to improve inspiratory muscle strength, prevent atrophy, and enhance ventilatory capacity [10,12].

This systematic review aimed to evaluate whether integrating IMT during the initial weaning trial improves extubation success compared to conventional strategies or no intervention. Additionally, it seeks to investigate the relationship between weaning success rates and parameters such as MIP and RSBI, providing evidence-based recommendations for incorporating IMT into clinical practice.

## 2. Materials and Methods

This systematic review adhered to the guidelines outlined in the Preferred Reporting Items for Systematic Reviews and Meta-Analyses (PRISMA) statement [13]. The protocol for this review was registered in a publicly accessible systematic review registry.

### 2.1. Eligibility Criteria

The review included randomized controlled trials (RCTs) that met specific inclusion criteria. Eligible studies involved participants aged 18 years or older who had been on mechanical ventilation (MV) for at least 48 h in an intensive care unit (ICU) due to respiratory failure. Studies involving COVID-19 patients or participants exclusively under 18 years of age were excluded [14].

The included studies compared IMT interventions, such as inspiratory resistive training, threshold pressure training, or ventilator-adjusted sensitivity, with conventional approaches like spontaneous breathing with a T-piece, standard physiotherapy, or the absence of specific treatments. Studies were required to report at least one of the following outcomes: weaning success, MIP, or the Rapid Shallow Breathing Index (RSBI).

### 2.2. Outcome Measures

The primary outcome measure was weaning success, defined as achieving at least 48 h of spontaneous breathing without the need of mechanical ventilation after successful extubation. Secondary outcome measures included changes in inspiratory muscle strength, assessed by MIP, and respiratory efficiency, evaluated using RSBI. These parameters were measured at the beginning and end of the weaning process.

### 2.3. Data Sources and Search Strategy

A comprehensive search of three electronic databases—PubMed, PEDro, and Cochrane—was conducted between April and August 2023, including articles published up to June 2023. No restrictions on the starting publication date were applied.

The search strategy was refined in three iterations. Initially, the term “ventilator weaning” yielded an overwhelming number of results (5807 on PubMed, 2439 on Cochrane, and 65 on PEDro). A second search combined “ventilator weaning” with “inspiratory muscle training”, which produced a more manageable number of records: 60 on PubMed, 62 on Cochrane, and 6 on PEDro. Finally, a third search added terms such as “spontaneous breathing” or “physical therapy modalities” to further narrow the scope, resulting in 33 articles on PubMed, 15 on Cochrane, and none on PEDro. Due to the limited yield of the third search, the second search results were used for study selection.

### 2.4. Data Screening and Extraction

Two independent reviewers (A.V.S. and N.W.) screened the titles and abstracts of the identified studies to determine eligibility based on the predefined criteria. Any discrepancies were resolved through discussion or by consulting a supervisor [15] for a final decision.

Full-text articles meeting the inclusion criteria were then independently reviewed by the two researchers. Data extracted included study design, participant demographics, details of the interventions and control conditions, and reported outcomes.

### 2.5. Quality Assessment

The risk of bias in the included RCTs was evaluated using the Cochrane Risk of Bias tool 2.0 (RoB 2.0) [16]. This tool assesses bias across several domains, including the randomization process, deviations from intended interventions, missing outcome data, outcome measurement, and selection of reported results. Each study was categorized as having a low risk of bias, some concerns, or a high risk of bias based on the evaluation criteria.

## 3. Results

A total of 128 studies were identified through the initial database search. After removing 22 duplicate records, 106 articles remained. Screening of titles and abstracts led to the exclusion of 39 studies that did not meet the inclusion criteria. An additional 48 studies were excluded because they did not utilize a randomized controlled trial (RCT) design. Of the remaining studies, 12 were excluded for specific reasons: one did not meet the eligibility criteria, eight were incomplete, and three lacked full-text availability. Ultimately, seven studies were deemed eligible for inclusion in this review (Figure 1).

### 3.1. Characteristics of the Included Studies

The seven studies collectively included 517 patients. Four studies [17,18,19,20] had a higher proportion of male participants, and the mean age across the studies was 67 years. All participants were adults aged 18 or older, and the primary indication for IMV was acute respiratory failure (ARF), often due to pneumonia or exacerbations of chronic obstructive pulmonary disease (COPD). Other cited reasons included sepsis, use of vasoactive drugs, and altered consciousness.

The interventions included IMT using an electronic Powerbreathe device (KH2 or K-5) (2/7) [20,21] or using a Threshold IMT device with a threshold adjusted to 20–50% of MIP (5/7) [17,18,19,22,23]. Comparisons included repeated spontaneous breathing with T-piece (2/7) [20,21], conventional physiotherapy management (4/7) [17,18,22,23], or no specific respiratory treatment (1/7) [19]. The outcomes that will be analyzed are weaning success (4/7) [18,19,20,22], MIP (7/7) [17,18,19,20,21,22,23], and RSBI (4/7) [17,20,22,23]. For a complete description of the included studies, see Table 1.

### 3.2. Risk of Bias of the Included Studies

In total, 7 studies were evaluated for the risk of bias using Cochrane’s tool named RoB 2.0 for randomised controlled trials. After assessing every outcome measure for every article, 3 articles were evaluated as presenting low-risk bias and 4 articles were assessed as showing some bias concerns. This indicates that 42.9% of the included studies show a low risk of bias. When analyzing the different domains, domain 3, representing the bias due to missing outcome data, is the only domain at low risk of bias for all the studies. In contrast, bias due to deviations from the intended intervention was the most common in the included studies.

The quality assessment scores using RoB 2.0 are shown in Table 2.

### 3.3. Effect of IMT on Weaning Success, MIP, and RSBI

Seven studies analyzed the effects of IMT on one or more of these outcome measures: weaning success, MIP, and RSBI. Three studies presented a low risk of bias (42.9%) [17,18,23] and the remaining four had some concerns (57.1%) [19,20,21,22].

Cader et al. [23] aimed to evaluate whether IMT improves MIP and RSBI in older intubated patients and improves weaning time. The intervention group received IMT for 5 min twice daily during the whole weaning period using a threshold device adjusted to 30% of the participants’ MIP. The aim was to increase it by 10% every day. MIP and RSBI were tested once every day in both groups before any IMT or other physiotherapy. Statistical difference was considered as *p* = 0.05. MIP increased significantly more in the intervention group compared to the control group (*p* < 0.00001). RSBI decreased in both groups, but change was significantly higher in the control group (*p* = 0.00259).

In 2013, Condessa et al. [17] published an RCT aiming to check if IMT accelerated weaning from IMV and studied the effects of IMT on MIP and RSBI. The intervention group was trained every day, 2 times a day, for 5 sets of 10 breaths. A threshold device was used for the IMT and adjusted to 40% of the participants’ MIP. MIP and RSBI were measured twice a day before every training session. Mean differences (95% CI) between groups were presented, and data was analyzed with a significance level of *p* < 0.05. The results show a significant increase in MIP in the intervention group compared to the control group (*p* = 0.000089). Even if it decreased in both groups, the difference in RSBI between them was statistically insignificant (*p* = 0.107).

The aim of Sandoval et al. [18] was to evaluate the efficacy of IMT in the weaning of IMV. Weaning success and MIP were analyzed in both groups. The intervention group was trained using a threshold IMT device adjusted to 50% of MIP in the initial stage. They were trained twice a day, every day, for 3 sets of 6 to 10 repetitions with 2 min rest between sets. Weaning success and MIP were analyzed, but no information on the measures’ timing and recurrence was given. An intention-to-treat analysis was performed, and the significance level was *p* < 0.25. The analysis of weaning success was based on logistic regression analysis, and the difference in the mean final change between groups was analyzed using the student *t*-test. There was no significant change in weaning success between groups (*p* = 0.54) and no statistical difference in the mean change of MIP between groups (*p* = 0.48).

Cader et al. [22] aimed to identify predictors of successful weaning and to evaluate the extubation process in the elderly receiving IMT. The intervention group underwent IMT using a threshold device and was trained twice a day for 5 min every day, starting at an initial load of 30% of MIP and increasing it by 10% daily. MIP and RSBI were measured during pre-testing and post-testing, just before extubation. Weaning success was also analyzed. The statistical significance level adopted was *p* < 0.05. A 2 × 2 analysis of a variance test was used to compare between groups and was followed by a post-hoc Tukey test. A Chi-square distribution analysis was used to evaluate weaning success. No significant difference in weaning success was found between the groups (*p* = 0.20). However, a significant difference in MIP (*p* = 0.001) and RSBI (*p* = 0.001) was found between groups.

The RCT published by Caruso et al. [19] tested whether adding IMT at the beginning of IMV would decrease the weaning process duration and describe the evolution of MIP with and without IMT. The intervention group was trained based on a trigger threshold adjusted to 20% of the first recorded MIP. Training started with a duration of 5 min and was increased by 5 min every day until reaching a total duration of 30 min. Afterwards, the load was increased by 10% of MIP. No information was mentioned about the significance level chosen, but it was reported that a linear regression of the daily measurements was used to assess the tendency of MIP. A negative coefficient showed a tendency for the MIP to decrease. No significant changes in MIP between groups (*p* = 0.342) was found, even though MIP values in the intervention group slightly decreased and those in the control group modestly increased.

In 2020, da Silva Guimaraes et al. [21] aimed to test if using IMT with an electronic resistive loading device could be associated with benefits in MIP, weaning, and survival rates. The intervention group underwent electronically assisted IMT (EIMT) using K-5 Powerbreathe daily in the morning from Monday to Friday. Weaning success and MIP were analyzed. MIP was measured at the beginning of the weaning process and repeated every 7 days until the end of IMV or death. Statistical significance was set at *p* < 0.05. The two-tailed T-test or the Mann–Whitney U test was used to examine differences between continuous variables. The Chi-square or Fisher exact test was used to determine the differences in categorical variables. The results showed a higher difference in the cumulative rate of weaning success (*p* = 0.001) and a significant increase in MIP (*p* = 0.003) in the intervention group compared to the control group.

Roceto Ratti et al. [20] aimed to compare EIMT, combined with spontaneous breathing, with T-piece and analyze the following variables: weaning success, MIP, and RSBI. The intervention consisted of IMT using the KH2 Powerbreathe to train twice daily for 3 sets of 10 repetitions with 1 min rest between sets. This group was subdivided into automatic (25/51) and manual (26/51) EIMT. MIP and RSBI were measured daily and compared to the baseline. Three measurements were taken, and only the most significant value was used in the analysis. The level of significance was set at *p* = 0.05. The differences between groups were examined using Chi-square, Pearson Chi-square, and Fisher exact tests. There was no significant difference in weaning success between groups (*p* = 0.45). However, MIP increased significantly in the control group (*p* < 0.01) and the automatic EIMT group (*p* = 0.007). RSBI decreased significantly in the control group (*p* = 0.03).

## 4. Discussion

The primary objective of this systematic review was to evaluate the evidence supporting the effectiveness of IMT in enhancing weaning success, respiratory muscle strength, and the ratio of RSBI. The findings indicate no significant differences between groups regarding weaning success. However, most studies reported a significant improvement in MIP within the intervention groups, while the results for RSBI were inconsistent across the included studies. Despite the positive correlation between IMT and RSBI, the lack of association between IMT and weaning success suggests that improvements in inspiratory muscle strength do not always directly translate into an effective transition to spontaneous breathing. This may be influenced by other factors, such as the functional reserve of the respiratory system, diaphragmatic fatigue, or the patient’s ability to sustain an efficient breathing pattern over time.

Despite the focus on assessing the impact of IMT during the early stages of the weaning process, the studies exhibited considerable heterogeneity in their interventions. Variations were observed in the devices, training frequencies, intensities, and load adjustments used for IMT, which complicates direct comparisons and highlights the lack of a standardized protocol for IMT interventions.

There were two types of interventions used in the included studies: threshold IMT adjusted to a pre-specified percentage of MIP and EIMT using Powerbreathe. Considering the threshold IMT [17,18,19,22,23], every article used a different percentage of MIP for the initial load of the device, ranging from 20% to 50% of the participants’ initial MIP measurement. In every article, the participants were exposed to different levels of training intensity. The problem encountered when comparing the studies using EIMT [20,21] was the use of two different devices. In addition, the training frequencies changed depending on the article. Some studies mentioned two training sessions per day, while others only trained once. For the duration of the training, it was expressed either in minutes or in sets with the respective repetitions per set. Finally, only two articles mentioned resting between sets. It demonstrates that no common baseline exists for the interventions performed in the included studies.

In four of the included studies, weaning success and MIP were analyzed as outcome measures [18,20,21,22]. In three of them [18,20,22], they found no significant between-group difference in weaning success. Cader et al. [22] and Roceto Ratti et al. [20] showed a significant increase in MIP in the intervention group, and Sandoval et al. [18] showed an insignificant increase in MIP between groups. Da Silva Guimaraes et al. [21] significantly improved weaning success and an increased MIP in the intervention group. Only one article shows the link between a significant increase in MIP and increased weaning success. In two of them, the increase in MIP has no impact on weaning success. However, Dos Santos Bien et al. demonstrated that an increased MIP correlates to ventilatory weaning success [25]. According to Bissett et al., [11] IMT during the first intubation stages only showed minimal benefits regarding weaning success. However, they also affirmed that the patients most likely to succeed with weaning from mechanical ventilation through IMT are those who failed the weaning the first time through other methods.

In two of the included studies, RSBI and weaning success were analyzed [20,22]. These studies showed no significant difference in weaning success between groups. One study reported a significant decrease in RSBI in the intervention group. However, another study found the opposite, with the control group presenting a significant decrease in RSBI compared to the intervention group. These findings demonstrate no correlation between decreased RSBI and weaning success rates. Other studies in the literature also reflect this inconsistency: while some describe RSBI as having moderate sensitivity and poor specificity for predicting extubation success, others consider it a useful predictor of successful weaning from mechanical ventilation [26]. However, this review does not support the latter conclusion.

### 4.1. Limitations

This systematic review has several limitations that should be acknowledged. One of the main challenges was the heterogeneity in the interventions and populations included in the studies. While all the selected articles investigated patients under IMV and IMT, the variations in training protocols, devices, and patient characteristics complicate the generalizability of the findings. For instance, the intensity, frequency, and duration of IMT interventions differed significantly, as did the methods used to measure outcomes such as MIP and the RSBI.

Additionally, weaning success, a critical outcome in this review, was inconsistently defined or measured across studies, further complicating comparative analysis. The limited number of studies focusing specifically on the relationship between IMT and weaning success restricted the ability to draw robust conclusions. Some studies did not account for confounding factors, such as the presence of comorbidities or variations in ICU protocols, which might have influenced the results.

Another limitation stems from the methodological quality of the included studies. Although efforts were made to include high-quality randomized controlled trials, some studies presented concerns regarding bias, particularly in the domains of deviations from intended interventions and selection of reported outcomes. These methodological discrepancies may have introduced variability in the results and affected the reliability of the findings.

Finally, the review did not include a meta-analysis due to the significant heterogeneity in the data, which limits the ability to provide quantitative synthesis and precise effect estimates. Future research should prioritize standardizing IMT protocols and outcome measures to enhance the comparability of studies and enable more definitive conclusions regarding the efficacy of IMT in improving weaning outcomes.

### 4.2. Future Directions

The qualitative analysis of the seven RCTs showed questionable results regarding the effectiveness of IMT on weaning success when comparing groups, a significant increase in MIP, and a significant decrease in RSBI in the intervention group in most of the articles. However, not all studies show the link between changes in MIP and RSBI and successful weaning. Homogenizing the type of IMT training and device used in the intervention group is also necessary. It is recommended that RCTs be developed using the same type of intervention and the same outcome measures. It is also recommended to collect information about muscle weakness in ICU patients or respiratory distress using values such as maximum intramuscular pressure or maximum respiratory flow. Future studies could also discriminate between patients presenting restrictive or obstructive pathophysiology. This would allow a more meaningful analysis of the studies.

## 5. Conclusions

The analysis of the included RCTs underscores the need for further research to clarify the role of IMT in weaning success. While IMT consistently demonstrated improvements in MIP, its impact on RSBI and weaning success remains inconclusive. Future studies should prioritize the standardization of IMT protocols, including training devices, load adjustments, frequency, and duration.

To achieve meaningful comparisons, future research should employ uniform outcome measures and clearly define key parameters such as weaning success. Large-scale, multicenter RCTs with rigorous methodologies are necessary to establish the relationship between changes in MIP, RSBI, and successful weaning from mechanical ventilation.

## Figures and Tables

**Figure 1 jfmk-10-00111-f001:**
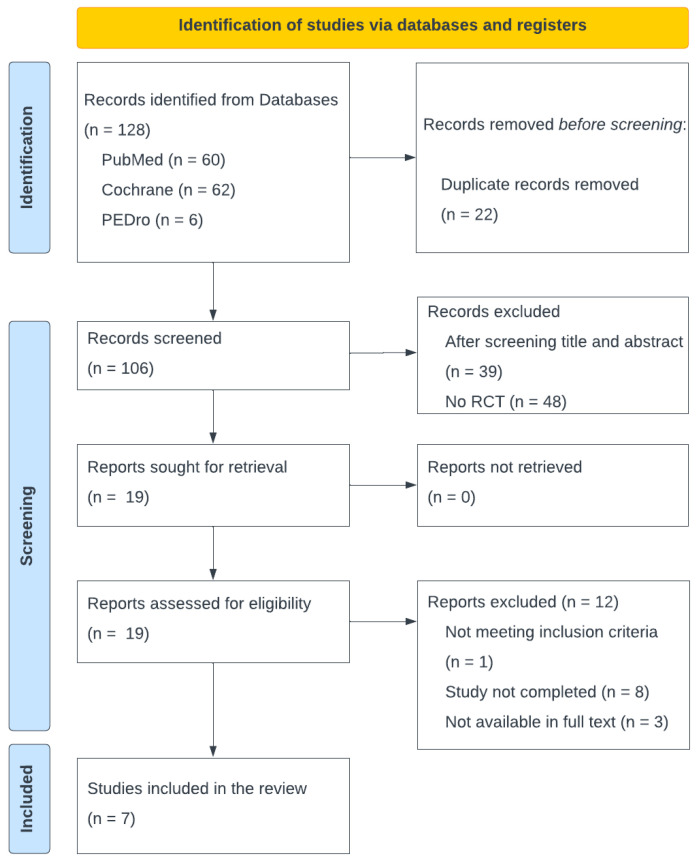
PRISMA flow diagram.

**Table 1 jfmk-10-00111-t001:** Characteristics of the included studies.

Author (Year)	Study Design	Population	Sample Size	Intervention	Control Group	Outcomes	Results
Cader et al. (2010) [23]	RCT	Intubated elderly (at least 70 years old), mechanically ventilated for at least 48 h in ICU	*n* = 41	*n* = 21 (57% F) Mean age: 83 ± 3 IMT using a threshold device with an initial load of 30% of MIP, 5 min 2×/day, everyday +usual care	*n* = 20 (50% F) Mean age: 82 ± 7 usual care only	MIP (vacuum manometer) RSBI (ventilometer)	MIP increased significantly more in the intervention group compared to the control group (*p* < 0.00001). RSBI decreased significantly more in the intervention group compared to the control group (*p* = 0.00259).
Condessa et al. (2013) [17]	RCT	Adults receiving pressure support ventilation for at least 48 h	*n* = 92	*n* = 45 (49% F) Mean age: 64 ± 17 IMT using threshold device with an initial load of 40% of MIP, 5 sets of 10 breaths 2×/day, every day +usual care	*n* = 47 (40% F) Mean age: 65 ± 15 Usual care only	MIP (vacuum manometer) RSBI (ventilometer)	Significant increase in MIP in the intervention group (*p* = 0.000089). No significant change in RSBI in both groups (*p* = 0.107).
Sandoval Moreno et al. (2019) [18]	RCT	Adults in mechanical ventilation for at least 48 h or more	*n* = 126 Mean age: 57	*n* = 62 (46.8% F) IMT with threshold IMT device adjusted to 50% of MIP, 3 series of 6–10 repetitions with 2 min rest between series, 2×/day, every day +conventional respiratory management	*n* = 64 (40.6% F) Conventional respiratory management	Weaning success MIP (digital manovacuometer)	No significant change in weaning success. (*p* = 0.54) No significant difference in the change of MIP between groups. (*p* = 0.48)
Cader et al. (2012) [22]	RCT	Elderly intubated in an ICU for at least 48 h	*n* = 28	*n* = 14 (57,14 F) Mean age: 82 ± 4 IMT with a threshold device initially adjusted to 30% of MIP, 5 min 2×/day, everyday +conventional physiotherapy	*n* = 14 (50% F) Mean age: 81 ± 6 Conventional physiotherapy	Weaning success MIP (vacuum manometer) RSBI (ventilometer)	No significant difference in weaning success between groups. (*p* = 0.20) MIP increased (*p* = 0.001), and RSBI decreased (*p* = 0.001) significantly in the intervention group compared to the control group.
Caruso et al. (2015) [19]	RCT	Critically ill adults, mechanically ventilated for at least 72 h	*n* = 25	*n* = 12 (33% F) Mean age: 67 ± 10 IMT performed by threshold device adjusted to 20% of MIP	*n* = 13 (31% F) Mean age: 66 ± 17 No respiratory muscle training	MIP (unidirectional valve)	Increase in MIP in the intervention group and a decrease in the control group, but not a significant (*p* = 0.34)
Da Silva Guimarães et al. (2020) [24]	RCT	Adults under mechanical ventilation for at least 48 h	*n* = 101	*n* = 48 (50% F) Mean age: 63 ± 16 IMT with EIMT with K-5 electronic Powerbreathe device + early mobilisation	*n* = 53 (53% F) Mean age: 69 ± 16 traditional T-piece protocol + early mobilisation	Weaning success MIP (digital vacuometer)	The intervention group had significantly better weaning success rates than the control group (*p* = 0.001). Significant increase in MIP in the intervention group compared to the control group (*p* = 0.003).
Roceto Ratti et al. (2022) [20]	RCT	Critically ill adults receiving invasive mechanical ventilation for at least 48 h	*n* = 104 (26% F) Mean age: 55 ± 17	*n* = 51 Subdivided into automatic EIMT (n = 25) and manual EIMT (n = 26) EIMT with KH2 electronic Powerbreathe device, 2x/day, every day, 3 sets of 10 repetitions with 1 min rest + usual care	*n* = 53 SB with T-piece + usual care	Weaning success MIP RSBI	No significant change in weaning success between groups (*p* = 0.45). Significant increase in MIP in both groups (*p* < 0.001). Significant decrease in RSBI in the control group (*p* = 0.03).

Abbreviations (in alphabetical order): EIMT (Electronically assisted Inspiratory Muscle Training); F (female); h (hours); ICU (Intensive Care Unit); IMT (Inspiratory Muscle Training); MIP (Maximal Inspiratory Pressure); RCT (Randomized Controlled Trial); RSBI (Rapid Shallow Breathing Index); SB (Spontaneous Breathing); × (times). The 7 studies were conducted in Brazil (6) and Columbia (1).

**Table 2 jfmk-10-00111-t002:** Risk of bias assessment of the RCTs using RoB 2.0.

Author (Year)	Random Sequence Generation	Deviations from the Intended Interventions	Missing Outcome Data	Measurement of Outcomes	Selection of the Reported Results	Overall Risk of Bias
Cader et al. (2010) [23]	Low risk	Low risk	Low risk	Low risk	Low risk	Low risk
Condessa et al. (2013) [13]	Low risk	Low risk	Low risk	Low risk	Low risk	Low risk
Sandoval Moreno et al. (2019) [18]	Low risk	Low risk	Low risk	Low risk	Low risk	Low risk
Cader et al. (2012) [22]	Low risk	Some concerns	Low risk	Low risk	Low risk	Some concerns
Caruso et al. (2005) [19]	Low risk	Some concerns	Low risk	Low risk	Some concerns	Some concerns
Da Silva Guimarães et al. (2020) [24]	Some concerns	Low risk	Low risk	Low risk	Low risk	Some concerns
Roceto Ratti et al. (2022) [20]	Low risk	Low risk	Low risk	Low risk	Low risk	Some concerns

## Data Availability

Data are contained within the article.

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
