# Peer review of "Inspiratory Muscle Training and Its Impact on Weaning Success in Mechanically Ventilated ICU Patients: A Systematic Review"

_jfmk, 2025, doi:10.3390/jfmk10020111_

Round 1
Reviewer 1 Report
Comments and Suggestions for Authors
Intresting article but really limited in the results and conclusions due to the heterogenicity of populations, techniques and threshold used. No information IS given if patiens had ICU weakness or how MIP were mesured. No inforamtion is given about respiratory encumbrance nore peak cough flow.
Pleas Define abbreviations first before using them, like in the abstract.
Author Response
RESPONSE TO REVIEWER 1
Comments and Suggestions for Authors
Intresting article but really limited in the results and conclusions due to the heterogenicity of populations, techniques and threshold used. No information IS given if patiens had ICU weakness or how MIP were mesured. No inforamtion is given about respiratory encumbrance nore peak cough flow.
RESPONSE: The findings of this review have highlighted the heterogeneity in the previous scientific literature. This has allowed us to identify key areas for improvement in future research conducted in these populations. We have included a section in the discussion on different lines of research to further investigate respiratory rehabilitation in the ICU. We suggest that future study designs should detail whether patients presented with muscle weakness or respiratory distress in the ICU, as well as information on maximum intramuscular pressure (MIP) or maximum cough flow (FCH).
Pleas Define abbreviations first before using them, like in the abstract.
RESPONSE: We have reviewed and corrected this aspect
Reviewer 2 Report
Comments and Suggestions for Authors
This is a very interesting review of the potential effect of respiratory muscle training on weaning outcome in mechanically ventilated patients in the ICU.
The main question addressed by the authors is the potential association between respiratory muscle training with respiratory physiotherapy and weaning outcome in ICU patients. In adition, the potential relation of IMP and different objective weaning readiness indices, such as RSBI. For these reasons, a systematic review of the literature was employed.
This topic is significant for all healthcare workers in the ICU, since early assessment of weaning readiness is very important in the ICU, takes a lot of time and increases daily burden for personell and patients. In addition, there are already conflicting data in the literature regarding impact of early or late respiratory physiotherapy on both weaning success and ICU mortality, in different groups of patients.
The authors have adequately shown the high eterogeneity of different protocols applyied across studies, as well as different groups of patients included
In this respect, I suggest that the authors sould further try to discriminate between patients with restrictive vs obstructive respiratory physiology and in addition, try to evaluate, if possible, clinical signs of weaning failure vs objective indices.
Conclusions of the systematic review are convincing and sound and highlight the great variability in terms of methods applyied in the literature for IMT, as well as heterogeneity of patients included in the studies. In addition, the authors should comment on the finding of lack of correlation between IMT and weaning outcome, despite positive correlations with RSBI. I suggest that other clinical indices of respiratory distress should be also analysed. Finally, maybe they should also try, if there are relevant data, to separate weaning from extubation failure, in order to reduce bias of studies as much as possible.
References, tables and figures are appropriate and clearly presented and illustrated.
Author Response
RESPONSE TO REVIEWER 2
Comments and Suggestions for Authors
This is a very interesting review of the potential effect of respiratory muscle training on weaning outcome in mechanically ventilated patients in the ICU.
The main question addressed by the authors is the potential association between respiratory muscle training with respiratory physiotherapy and weaning outcome in ICU patients. In adition, the potential relation of IMP and different objective weaning readiness indices, such as RSBI. For these reasons, a systematic review of the literature was employed.
RESPONSE: Thank you for appreciating our work
This topic is significant for all healthcare workers in the ICU, since early assessment of weaning readiness is very important in the ICU, takes a lot of time and increases daily burden for personell and patients. In addition, there are already conflicting data in the literature regarding impact of early or late respiratory physiotherapy on both weaning success and ICU mortality, in different groups of patients.
RESPONSE: Thank you very much for your comment
The authors have adequately shown the high eterogeneity of different protocols applyied across studies, as well as different groups of patients included
In this respect, I suggest that the authors sould further try to discriminate between patients with restrictive vs obstructive respiratory physiology and in addition, try to evaluate, if possible, clinical signs of weaning failure vs objective indices.
RESPONSE: With the current data, this distinction is not possible. We have added a sentence to the discussion section on future lines of research: "Future studies could also distinguish between patients with restrictive or obstrictive pathophysiology."
Conclusions of the systematic review are convincing and sound and highlight the great variability in terms of methods applyied in the literature for IMT, as well as heterogeneity of patients included in the studies. In addition, the authors should comment on the finding of lack of correlation between IMT and weaning outcome, despite positive correlations with RSBI.
RESPONSE: We add in discussion: “Despite the positive correlation between IMT and RSBI, the lack of association between IMT and weaning success suggests that improvements in inspiratory muscle strength do not always directly translate into an effective transition to spontaneous breathing. This may be influenced by other factors, such as the functional reserve of the respiratory system, diaphragmatic fatigue, or the patient’s ability to sustain an efficient breathing pattern over time.”
I suggest that other clinical indices of respiratory distress should be also analysed. Finally, maybe they should also try, if there are relevant data, to separate weaning from extubation failure, in order to reduce bias of studies as much as possible.
RESPONSE: Thanks for your comment. With the current data, it is not possible to delve deeper into this aspect. However, we have mentioned it in the section on future lines of research to suggest ways to advance our knowledge in subsequent clinical studies.
References, tables and figures are appropriate and clearly presented and illustrated.
RESPONSE: Thank you for reviewing this section.